# Long-Term Consumption of Sucralose Induces Hepatic Insulin Resistance through an Extracellular Signal-Regulated Kinase 1/2-Dependent Pathway

**DOI:** 10.3390/nu15122814

**Published:** 2023-06-20

**Authors:** Meng-Jie Tsai, Chung-Hao Li, Hung-Tsung Wu, Hsin-Yu Kuo, Chung-Teng Wang, Hsiu-Ling Pai, Chih-Jen Chang, Horng-Yih Ou

**Affiliations:** 1Department of Internal Medicine, National Cheng Kung University Hospital, College of Medicine, National Cheng Kung University, Tainan 70403, Taiwan; n103264@mail.hosp.ncku.edu.tw (M.-J.T.); telomere-aging@hotmail.com.tw (H.-Y.K.); 2Department of Family Medicine, An Nan Hospital, China Medical University, Tainan 70965, Taiwan; smallhear@gmail.com; 3School of Medicine, College of Medicine, China Medical University, Taichung 40402, Taiwan; 4Department of Internal Medicine, School of Medicine, College of Medicine, National Cheng Kung University, Tainan 70101, Taiwan; z11008014@ncku.edu.tw (H.-T.W.); knight790105@hotmail.com (C.-T.W.); 5Graduated Institute of Metabolism and Obesity Science, College of Nutrition, Taipei Medical University, Taipei City 11031, Taiwan; hsiuuuoo7635@gmail.com; 6Department of Family Medicine, Ditmanson Medical Foundation Chia-Yi Christian Hospital, Chiayi City 60002, Taiwan

**Keywords:** artificial sweetener, high-fat diet, insulin resistance, sucralose, Taste 1 receptor member 3

## Abstract

Sugar substitutes have been recommended to be used for weight and glycemic control. However, numerous studies indicate that consumption of artificial sweeteners exerts adverse effects on glycemic homeostasis. Although sucralose is among the most extensively utilized sweeteners in food products, the effects and detailed mechanisms of sucralose on insulin sensitivity remain ambiguous. In this study, we found that bolus administration of sucralose by oral gavage enhanced insulin secretion to decrease plasma glucose levels in mice. In addition, mice were randomly allocated into three groups, chow diet, high-fat diet (HFD), and HFD supplemented with sucralose (HFSUC), to investigate the effects of long-term consumption of sucralose on glucose homeostasis. In contrast to the effects of sucralose with bolus administration, the supplement of sucralose augmented HFD-induced insulin resistance and glucose intolerance, determined by glucose and insulin tolerance tests. In addition, we found that administration of extracellular signal-regulated kinase (ERK)-1/2 inhibitor reversed the effects of sucralose on glucose intolerance and insulin resistance in mice. Moreover, blockade of taste receptor type 1 member 3 (T1R3) by lactisole or pretreatment of endoplasmic reticulum stress inhibitors diminished sucralose-induced insulin resistance in HepG2 cells. Taken together, sucralose augmented HFD-induced insulin resistance in mice, and interrupted insulin signals through a T1R3-ERK1/2-dependent pathway in the liver.

## 1. Introduction

The Western diet has become increasingly prevalent in modern society, leading to excessive calorie intake and high sugar consumption. This dietary shift has contributed to a rise in obesity rates. Obesity has a strong correlation with insulin resistance and is associated with an elevated risk of developing diabetes mellitus, coronary heart disease, cerebrovascular events, and non-alcoholic fatty liver disease (NAFLD) [1,2]. Addressing the battle against obesity has become a significant challenge in recent years.

In the 1800s, artificial sweeteners were first introduced into the food industry [3]. Non-nutritive artificial sweeteners, which have minimal to no calories, quickly gained popularity as sugar substitutes among individuals with diabetes or obesity who sought to manage their body weights and maintain glucose homeostasis [4,5]. A survey conducted between 2009 and 2012 revealed that over one-quarter of children and more than 40% of adults consumed low-calorie sweeteners in the United States [6].

Following the increasing popularity of artificial sweeteners over the years, a number of recent studies investigated the possible adverse effects of artificial sweeteners. Studies have established that ingestion of non-nutritive sweeteners alters the metabolome in blood and disrupts the balance of the gut and oral microbiome, leading to dysbiosis and impairing the downstream glycemic response, resulting in glucose intolerance [7,8]. Moreover, the intake of acesulfame potassium (AceK) has been found to increase atherosclerotic plaque formation, promote hepatic lipogenesis, and exacerbate dyslipidemia induced by a high-cholesterol diet in apolipoprotein E knockout mice [9]. In addition, the usage of commonly used artificial sweeteners, such as sucralose, AceK, and aspartame has been connected with a heightened likelihood of cardiovascular disorders, including coronary heart disease and cerebrovascular disease [10]. Furthermore, consumption of aspartame in pregnant mice brings a negative impact on fetal development and placenta growth. In addition, trophoblasts treated with aspartame exhibit cell cycle cessation and impaired cell proliferation, owing to increased oxidative stress [11]. These observations have raised concerns about the use of artificial sweeteners.

Among these commonly used artificial sweeteners, sucralose is a non-nutritive artificial sweetener which is a derivative of sucrose with the substitution of three hydroxyl groups by three chlorine atoms. Sucralose possesses a sweetness intensity that is approximately 600-fold greater than that of regular sugar [12]. It is authorized worldwide and is added to various food products and beverages. Although sucralose is extensively used, a variety of adverse effects on the human body were reported. It was known that high dose of sucralose intake hinders T cell proliferation and differentiation in mice, altering the structure of the T cell membrane and dampening cell signaling. This has resulted in diminished CD8^+^ T cell response in cancer and bacterial infection models, as well as a decrease in T cell-dependent autoimmunity [13]. Furthermore, sucralose intake increases lipid peroxidation and triggers oxidative stress responses, disrupting the embryonic development of *Danio rerio* (zebra fish). This disruption has been linked to various malformations, such as scoliosis, yolk deformation, craniofacial malformations, delayed hatching, and even mortality [14]. Regarding metabolic effects, research on mice indicates that maternal consumption of sucralose can trigger gut inflammation, hinder intestinal development, and disrupt the barrier function in offspring. These changes alter fatty acid biosynthesis and metabolism, resulting in gut dysbiosis and exacerbating hepatic steatosis when exposed to a high-fat diet (HFD) [15]. In addition, sucralose activates taste receptor type 1 member 3 (T1R3), generating reactive oxygen species (ROS), and triggering endoplasmic reticulum (ER) stress and lipogenesis, further accelerating the development of hepatic steatosis [16]. Nevertheless, the precise effects of sucralose on glucose metabolism and the underlying mechanisms are largely unknown.

In this study, we not only investigated the effects of short-term sucralose intake on glucose utilization in mice, but also examined the impact of long-term consumption of sucralose on blood glucose regulation. We closely scrutinized the changes in glycemic parameters in mice fed a high-fat diet with supplement of sucralose (HFSUC). Additionally, we employed HepG2 cell models and utilized inhibitors for various enzymes to clarify the underlying mechanisms of how sucralose affects glucose homeostasis.

## 2. Materials and Methods

### 2.1. Animals

Approval for all animal experiments was granted by the Institutional Animal Care and Use Committee of Taipei Medical University (IACUC No: LAC-2020-0302) and the Institutional Animal Care and Use Committee of National Cheng Kung University (IACUC No: 111–209). All the animal experiments were conducted in accordance with the “Guide for the Care and Use of Laboratory Animals” provided by the National Research Council (US) Committee. Six-week-old C57BL/6 male mice were purchased from National Laboratory Animal Center (Taipei, Taiwan), and housed in the environment with a temperature of 23 ± 2 °C, humidity 60 ± 10%, and alternative light and dark every 12 h (lights on at 07:00 a.m.), without restriction to food or water. When the mice were at eight weeks old, they were fed with a normal chow diet (Laboratory Rodent Diet #5001, LabDiet St. Louis, MO, USA), HFD (58Y1, 60% kcal from fat, TestDiet, St. Louis, MO, USA) or a HFD supplemented with 0.06% sucralose (Alfa Aesar, Ward Hill, MA, USA) for two weeks. The amount of food intake and water consumption was measured using metabolic cages. The U.S. Food and Drug Administration provided the acceptable daily intake (ADI) of sucralose as 5 mg/kg/day [17]. Therefore, we converted the dose from human to mice based on body surface area [18]. The acceptable daily intake of sucralose for mice was 60 mg/kg/day, which is equivalent to 600 ppm in feed, the same as the supplemented 0.06% sucralose in the HFD used in this study. Furthermore, intraperitoneal injections of U0126 at a dosage of 10 mg/kg/day [19] in the mice of the HFSUC group for two weeks was applied to evaluate the role of ERK1/2 in sucralose-induced insulin resistance.

### 2.2. Single Dose of Sucralose Supplement via Oral Gavage

The chow-fed mice received a fasting period of 15 h. Blood samples were collected at indicated time points of the mice after administering a sucralose solution of 1 g/kg body weight via oral gavage. The concentrations of plasma glucose were measured using a glucose meter (Accu-Chek Performa, Roche, Basel, Switzerland). Commercial insulin enzyme-linked immunosorbent assay kits (Mercodia, Uppsala, Sweden) were utilized for plasma insulin concentrations measurement.

### 2.3. Glucose and Insulin Tolerance Tests

Each group of mice received a fasting period of 6 h before the experiment. After administering glucose (1 g/kg body weight) via oral gavage or insulin injection intraperitoneally (1 U/kg body weight), blood samples were collected at indicated time points for the measurement of plasma glucose concentrations [20,21].

### 2.4. Determination of Hepatic Insulin Sensitivity In Vivo

Each group of mice was fasted for 24 h and well anesthetized. The animals then received an injection of recombinant insulin protein (Humulin R^®^, Eli Lilly and Company, Indianapolis, IN, USA) at a dose of 5 units through the portal vein. After five minutes of the injection, the liver tissues were removed for further analysis.

### 2.5. Cell Culture

The HepG2 cell line was purchased from Bioresource Collection and Research Center (Food Industry Research and Development Institute, Hsinchu, Taiwan) and maintained (5% CO_2_, 37 °C) in Dulbecco’s modified Eagle medium (DMEM, HyClone, South Logan, UT, USA) provided with 10% heat-inactivated fetal bovine serum. The cells were seeded at a density of 2 × 10^5^/well on a 6 cm diameter cell culture dish and maintained overnight in low-glucose DMEM without serum for subsequent experiments. The cells were pretreated with various doses of sucralose from 0.1 to 10 mM for 30 min, followed by treatment with 1 μM insulin for an additional 30 min. Protein lysates were collected for the determination of Akt phosphorylation using Western blots. In addition, the cells were treated with 10 mM sucralose for indicated times and the protein lysates were collected to determine the expressions of mitogen-activated protein kinases (MAPK) by Western blots. Furthermore, the cells were pretreated with JNK1/2 inhibitor (SP600125), ERK1/2 inhibitor (U0126), inositol-requiring enzyme type 1 (IRE1) inhibitor (STF083010) (Cell Signaling, Danvers, MA, USA), taste receptor type 1 member 2 (T1R2) inhibitor (Gymnemic acid I, Taiclone, Taipei, Taiwan), or T1R3 inhibitor (Lactisole, Cayman, MI, USA) at indicated doses for 30 min, respectively. Then, they were treated with 10 mM sucralose for 30 min followed by 1 µM insulin for another 30 min. Protein lysates for each group of the experiment were collected for Western blot analysis.

### 2.6. Western Blot Analysis

The protein samples from liver tissues or HepG2 cells were extracted and mixed with radioimmunoprecipitation lysis buffer (VWR Chemical solon, Solon, OH, USA) containing protease inhibitors (Sigma-Aldrich, St. Louis, MO, USA). Following centrifugation at 13,000× *g* rpm at 4 °C for 10 min, the liquid portion was separated, and the protein content was assessed by utilizing a bicinchoninic acid assay kit (Visual Protein, Taipei, Taiwan). The proteins were separated by 10% sodium dodecyl sulfate-polyacrylamide gel electrophoresis (SDS-PAGE), and transferred to a polyvinylidene difluoride membrane (Biomate, Taipei, Taiwan). At room temperature, the membranes were treated with 10% skim milk for 1 h. Subsequently, they were incubated with primary antibodies diluted at a ratio of 1:1000, including phospho-Akt (pAkt), Akt, phospho-ERK1/2 (pERK1/2), ERK1/2, phospho-p38 (pP38), p38, phospho-JNK1/2 (pJNK1/2), and JNK1/2 (Cell Signaling, Danvers, MA, USA) at 4 °C overnight. Following the rinsing of the membranes with a solution containing 10 mM Tris (pH 7.6), 150 mM NaCl, and 0.05% Tween 20, the blots were subjected to incubation at room temperature for 1 h with secondary antibodies conjugated to horseradish peroxidase, diluted at a ratio of 1:5000. The protein bands were detected using Immobilon (Millipore, Billerica, MA, USA). Further, the signal intensity was quantified by ImageJ software.

### 2.7. Statistics

Illustration and statistical analyses were conducted by utilizing GraphPad Prism 8. The data were showcased using the mean ± standard error (SEM). Student’s *t*-test or one-way ANOVA followed by Tukey’s post hoc test was used. Statistical significance was defined as the *p*-value was less than 0.05.

## 3. Results

### 3.1. Oral Bolus Administration of Sucralose Enhances Plasma Insulin Level to Decrease Plasma Glucose Level

With the aim of clarifying the effects of short-term administration of sucralose on blood glucose levels, mice fed with normal chow were given a single dose of sucralose solution via oral gavage. We found that after treatment of sucralose, the plasma concentrations of insulin in the mice were significantly increased after fifteen minutes (Figure 1A), while the concentration of plasma glucose significantly decreased simultaneously (Figure 1B). These findings indicate that bolus administration of sucralose may stimulate insulin secretion, leading to a reduction in plasma glucose levels.

### 3.2. Supplement of Sucralose Exhibited No Significant Effects on Body Weight and Food Intake in Mice Fed with HFD

In addition to studying the short-term influence of sucralose consumption, we conducted further investigations to examine the long-term effects of sucralose over a two-week duration. As shown in Figure 2, the body weights of the mice were significantly increased in both the HFD group (25.3 ± 0.2 g; *p* < 0.001) and the HFSUC group (25.8 ± 0.6 g; *p* < 0.01), as compared with the Chow group (23.1 ± 0.2 g). Nevertheless, there were no significant differences between the body weight of the HFSUC group and HFD group (Figure 2A). In contrast, there were no significant differences among the three groups in terms of daily calorie intake (Chow, 23.1 ± 1.2 kcal; HFD, 19.9 ± 0.8 kcal; HFSUC, 20.5 ± 0.8 kcal/day) (Figure 2B). However, the water intake was significantly decreased in both the HFD group (3.4 ± 0.1 mL; *p* < 0.001) and the HFSUC group (2.9 ± 0.1 mL/day; *p* < 0.001), as compared with the Chow group (5.6 ± 0.2 mL) (Figure 2C).

### 3.3. Long-Term Administration of Sucralose Augmented HFD-Induced Glucose Intolerance and Insulin Resistance

For the purpose of investigating the effects of sucralose on glucose homeostasis in HFD mice, oral glucose tolerance and insulin tolerance tests were conducted. The fasting glucose levels exhibited no significant differences among the Chow group (154.9 ± 11.5 mg/dL), the HFD group (169.5 ± 6.6 mg/dL), and the HFSUC group (169.6 ± 7.7 mg/dL) (Figure 3A). However, as shown in Figure 3B, blood glucose levels significantly increased after 30 min in the oral glucose tolerance test as compared with the Chow group (*p* < 0.01), indicating impaired glucose tolerance in the HFD group. Moreover, the supplement of sucralose in the HFD group augmented the effects of HFD on glucose intolerance (*p* < 0.05). The worsened glucose utilization was confirmed by the area under curve (AUC) in the HFSUC group after administration of sucralose (Figure 3C). We further investigated the effects of sucralose on insulin sensitivity in mice. As shown in Figure 3D, insulin sensitivity, validated by the insulin tolerance test, was substantially decreased in the HFD group, as compared with the Chow group (*p* < 0.01). Likewise, decreased insulin sensitivity was substantiated by AUC in the insulin tolerance test after the sucralose supplement (Figure 3E).

To confirm the augmentation of HFD-induced insulin resistance in HFSUC mice, hepatic insulin signals were evaluated. Decreased pAkt signals were found in the HFD group, as compared with the Chow group (*p* < 0.05), indicating impaired insulin signaling and compromised glucose utilization by peripheral tissues or liver cells [22]. After the supplement of sucralose in the HFD group, decreased pAkt signals were observed (*p* < 0.001) (Figure 3F). These findings imply that consumption of sucralose may exacerbate HFD-induced insulin resistance, leading to impaired glucose homeostasis.

### 3.4. Sucralose Impaired Insulin Signals through an ERK1/2-Dependent Pathway

In order to investigate the potential mechanisms of sucralose-induced insulin resistance, we utilized the HepG2 cell line. This cell line was chosen due to its widespread availability and its capability to express insulin signaling [23]. As shown in Figure 4A, treatment of sucralose dose-dependently suppressed insulin-induced Akt phosphorylation, indicating the development of insulin resistance in HepG2 cells. It was known that elevated ERK activity is correlated with increased insulin resistance [24], and sucralose has an activity to stimulate ERK1/2 [25]. We then investigated the role of ERK1/2 in sucralose-induced insulin resistance. Treatment with sucralose led to a dose-dependent increased phosphorylation of MAPK, particularly ERK1/2 (Figure 4B) within five minutes, and JNK1/2 (Figure 4C) within 1 h in HepG2 cells. However, there were no significant changes observed in the phosphorylation of P38 after sucralose treatment (Figure 4D). Pretreatment of U0126, an ERK inhibitor, reversed the effect of sucralose on the phosphorylation of Akt (Figure 4E). However, SP600125, a JNK inhibitor, revealed no significant effects on sucralose-induced insulin resistance in HepG2 cells (Figure 4F).

Regarding the impact of ERK1/2 on the glucose homeostasis in vivo, we found that the activity of ERK1/2 was increased in mice fed with HFD as compared with the Chow group. Furthermore, we observed that the supplement of sucralose augmented the effects of HFD on the phosphorylation of ERK1/2 (Figure 5A). In addition, administration of U0126 resulted in improved glucose intolerance during the OGTT in HFSUC mice. This improvement in glucose utilization was further validated by the AUC following U0126 administration (Figure 5B,C). In ITT, administration of U0126 led to a decrement in blood glucose levels in HFSUC mice (Figure 5D). Additionally, improved insulin sensitivity was confirmed by the AUC after administering U0126 (Figure 5E). These findings indicate that sucralose intake induced insulin resistance through an ERK1/2-dependent pathway.

### 3.5. Sucralose Induces Insulin Resistance through IRE1α and T1R3

It has been established that sucralose increases ER stress in various cells [16,26], and ER stress is closely connected to the development of insulin resistance [27]. To explore the impact of ER stress on sucralose-induced insulin resistance, HepG2 cells were treated with STF083010, an ER stress-related IRE1α inhibitor. According to the information depicted in Figure 6A, pretreatment with STF083010 reversed the effects of sucralose on the expression of pAkt dose dependently, indicating ER stress was mediated in sucralose-induced insulin resistance.

Previous studies have found that sucralose is a ligand of sweet taste receptors [28], and stimulation of sweet taste receptors is related to an increase in oxidative stress [11,29]. In the present study, pretreatment of gymnemic acid (GA), a T1R2 inhibitor revealed no significant effects of sucralose on Akt phosphorylation (Figure 6B). However, lactisole (LAC), a T1R3 inhibitor, reversed the effects of sucralose on pAkt expression in a dose-dependent manner, indicating that sucralose enhances insulin resistance through T1R3 (Figure 6C).

## 4. Discussion

Sucralose, a commonly used artificial sweetener, has been reported to potentially disrupt glycemic homeostasis, yet the underlying mechanisms are not fully established. To our understanding, this study represents the initial investigation targeting the underlying pathways of sucralose-induced insulin resistance in animal and cell models. Our research indicates that sucralose treatment impaired insulin sensitivity and glucose utilization in both mice and HepG2 cells, shedding light on the detrimental influence of sucralose on glucose regulation.

Previous studies indicate that sucralose consumption offers metabolic benefits, such as promoting weight loss in humans and reducing feelings of hunger [30,31]. It triggers the activation of sweet taste receptors, specifically T1R2 and T1R3, which are present in taste cells located on the lingual epithelium of the tongue and in enteroendocrine cells within the intestines. These receptors have a critical influence in detecting the presence of sugars or sweeteners within the gastrointestinal tract [32]. Sucralose also stimulates the release of glucagon-like peptide-1 (GLP-1), a hormone that serves a crucial role in glycemic regulation. This effect has been observed in healthy individuals, where sucralose intake helps decrement in blood glucose levels [33,34,35]. Furthermore, sucralose has been found to upregulate the expression of glucose transporters and sweet taste receptors in the intestines, resulting in a reduction in blood glucose levels [36]. In addition, in comparison to mice fed with sucrose, those fed with sucralose at doses within the ADI range exhibited decreased fat accumulation, improved plasma low density lipoprotein-cholesterol levels, reduced hepatic lipid deposition, and enhanced glucose tolerance [37].

On the contrary, multiple studies have reported various detrimental metabolic effects linked to the consumption of sucralose. For example, even at a dose as low as 15% of the ADI, sucralose has been found to impair insulin sensitivity in healthy individuals [38]. Moreover, the consumption of sucralose has been demonstrated to alter the composition of gut microbiota and its metabolites. Specifically, it promotes the growth of *Bacteroides* and *Clostridium* species, which are responsible for the production of deoxycholic acid (DCA). This increase in deoxycholic acid levels can be observed in various biological matrixes, including liver, serum, and feces of mice. Consequently, elevated hepatic deoxycholic acid levels may disrupt gene expression in the liver and contribute to the development of sucralose-induced NAFLD in mice [39]. Sucralose consumption also exacerbates HFD-induced hepatic steatosis [16]. Therefore, the effects of sucralose on metabolism remain controversial and the precise mechanisms are still obscure. In this study, we found an initial elevation in insulin secretion and a subsequent reduction in plasma glucose levels after administering a single dose of sucralose to mice. However, when sucralose was administered for a longer duration of two weeks, we observed a deterioration in glucose utilization and insulin sensitivity in mice fed with HFD, as confirmed by the oral glucose tolerance test and the insulin tolerance test. It is worth noting that there were no significant differences observed in daily calorie intake among the Chow group, the HFD group, and the HFSUC group. These findings indicate that long-term sucralose usage may contribute to impaired glucose regulation as well as insulin sensitivity, regardless of caloric intake.

MAPK signals and PI3K/Akt pathways are closely linked to the regulation of insulin sensitivity and have a significant impact on in the pathophysiology of type 2 diabetes mellitus [40,41]. Activation of ERK1/2 signaling pathway contributes to a decrease in adiponectin expression and an increase in lipolysis activity. Consequently, the released free fatty acids from lipolysis can enhance the expression of inflammatory cytokines and provoke an inflammatory response, potentially contributing to the development of insulin resistance [24]. JNK1/2, another component of the MAPK family, has a substantial role on disrupting insulin action and exacerbating insulin resistance. The activation of JNK1/2 can be triggered by heightened levels of free fatty acids as well as inflammatory markers, including TNF-α. Notably, JNK1/2 activity is abnormally increased in individuals with obesity [42]. Furthermore, JNK1/2 contributes to insulin resistance through various mechanisms, such as phosphorylation of insulin receptor substrates (IRS)-1/2 and the stimulation of inflammation in metabolic processes. Inhibition of JNK1/2 is considered as a potential strategy to alleviate both obesity and insulin resistance [43]. Additionally, p38/MAPK signaling plays pivotal role in promoting adipose tissue inflammation and is significantly involved in HFD-induced obesity. Moreover, deletion of p38 has been shown to improve hepatic steatosis, even under conditions of HFD treatment [44]. In addition to MAPK, ER stress has been acknowledged as a vital factor in the progression of adverse metabolic outcomes associated with obesity as well as type 2 diabetes [45]. This heightened ER stress can result in an increase in proinflammatory cytokines, for example, TNF-α and interleukin 6, which further exacerbate inflammation [46]. Additionally, it disrupts the insulin signaling pathway, thus contributing to the development of insulin resistance [47]. The unfolded protein response (UPR) is controlled by three transmembrane mediators as well as ER stress sensors, including IRE1α, RNA-dependent protein kinase (PKR)-like ER kinase (PERK), and activating transcription factor 6 (ATF6) [48]. Among these, IRE1 is particularly prominent and has been implicated in the regulation of pancreatic beta cell function. In pathological conditions, heightened ER stress can intensify the activation of IRE1, leading to disruption in the balance of insulin requirements. This dysregulation may have significant effects on glycemic homeostasis [49]. In accordance with these studies, we found that sucralose consumption decreases insulin signal transduction in both HepG2 cells and mice liver tissues. Additionally, sucralose activated the ERK1/2 and JNK1/2 pathways. Interestingly, while the JNK1/2 inhibitor SP600125 did not reverse sucralose-induced impaired insulin signaling, U0126, an ERK1/2 inhibitor dose-dependently improved insulin sensitivity in HepG2 cells. Concomitant administration of U0126 with HFSUC mice also demonstrates improvement in glucose utilization and insulin sensitivity. These results suggest that sucralose does not affect plasma glucose levels through the JNK1/2 pathway, but rather through the ERK1/2 pathway. Furthermore, the inhibition of IRE1α and T1R3 effectively reversed the interruption of insulin signal transduction caused by sucralose treatment in HepG2 cells. These findings indicate that activation of IRE1 and T1R3 may have a significant effect on mediating sucralose-induced insulin resistance.

In contrast to artificial sweeteners, some natural sweeteners show beneficial effects on human health. Stevioside, derived from *Stevia rebaudiana*, is a natural sweetening agent known for its sweetness intensity, which ranges from 150 to 300 times greater than that of sucrose [50]. Extensive research has not only confirmed its safety but also revealed numerous health benefits [51]. Studies have established that stevioside possesses the capacity to stimulate peripheral μ-opioid receptors, leading to a reduction in plasma glucose levels and promoting hepatic glycogen synthesis [52]. Additionally, stevioside has been found to increase insulin levels and improve insulin sensitivity, making it beneficial for individuals with diabetes [53,54]. Moreover, it has been observed to lower blood pressure in individuals with mild essential hypertension [55]. Further, stevioside demonstrated a potential in weight control [56], controlling dyslipidemia [57], and inhibiting the formation of atherosclerotic plaques [58], which contribute to cardiovascular health. It also exhibits antioxidant properties, neutralizing free radicals and reducing their damaging effects [59]. Additionally, preliminary research suggests that stevioside may have anticancer properties, as it has been found to decrease the viability of cancer cells [60]. Conversely, excessive consumption of licorice, a commonly used sweetener, can lead to serious complications. The active metabolite in licorice, glycyrrhizic acid, inhibits the activity of 11β-hydroxysteroid dehydrogenase type 2, leading to an elevated plasma cortisol levels [61]. This, in turn, stimulates mineralocorticoid receptors in the renal distal tubules, resulting in the development of apparent mineralocorticoid excess (AME) syndrome. AME is characterized by refractory hypokalemia, hypertension, metabolic alkalosis, and chronic ingestion of licorice may have unfavorable effects on the human body [62]. Further, sorbitol is a low-calorie sweetener found in various fruits. It can also be produced endogenously in the human body under hyperglycemic conditions [63]. However, long-term ingestion of sorbitol has been found to alter the composition of the gut microbiome in mice and result in glucose intolerance [64]. Additionally, the accumulation of sorbitol can contribute to multiple complications associated with diabetes [65]. Therefore, additional investigations are required to determine whether natural sweeteners offer a more favorable alternative to artificial sweeteners.

In summary, the results of our investigation indicate that prolonged consumption of sucralose exacerbates HFD-induced insulin resistance in mice by disrupting insulin signal transduction through the involvement of T1R3 and IRE1α. Moreover, besides the frequency and duration of consumption, the dosage of sucralose exposure plays a crucial role in its adverse effects. Although the doses used in our study were within the ADI limit for humans, the addition of sucralose still impaired insulin sensitivity and disrupted glucose homeostasis. Therefore, it may be necessary to reassess and potentially adjust the ADI of sucralose to mitigate its adverse effects. Further investigations are warranted to gain a comprehensive understanding of the potential risks associated with sucralose consumption and to establish appropriate guidelines for its safe use.

## Figures and Tables

**Figure 1 nutrients-15-02814-f001:**
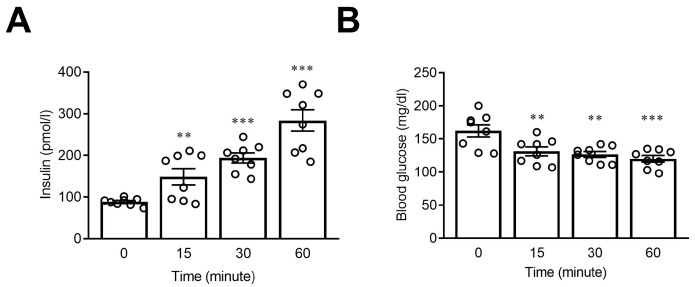
Short-term administration of sucralose increased plasma insulin concentration to decrease plasma glucose levels in mice. Blood samples were collected at indicated time points to determine plasma insulin (**A**) and glucose (**B**) concentrations. *n* = 8 for each group of mice. ** *p* < 0.01, and *** *p* < 0.001, as compared with the data before sucralose administration (0 min).

**Figure 2 nutrients-15-02814-f002:**
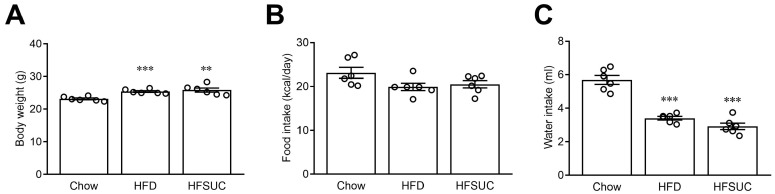
The effects of long-term consumption of sucralose on body weight, food, and water intake in mice. Eight-week-old C57BL/6 mice were fed with chow diet (Chow), high-fat diet (HFD) or high-fat diet supplemented with sucralose (HFSUC) for two weeks. The body weight (**A**), daily calorie intake (**B**), and amount of water intake (**C**) were measured in each group. *n* = 6 for each group of the mice. ** *p* < 0.01, and *** *p* < 0.001, as compared with the Chow group.

**Figure 3 nutrients-15-02814-f003:**
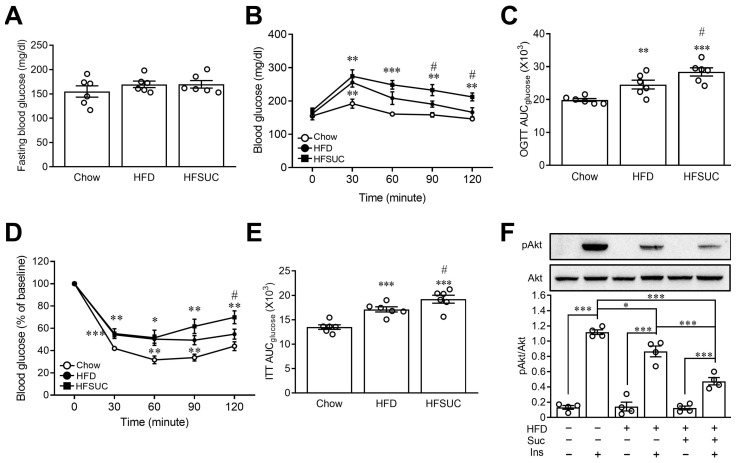
Long-term administration of sucralose provoked high-fat diet-induced insulin resistance in mice. The fasting blood glucose levels were measured in mice fed with chow diet (Chow), high-fat diet (HFD) or HFD supplemented with sucralose (HFSUC) for two weeks (**A**). The plasma glucose levels were measured at indicated times in the oral glucose tolerance test (OGTT) (**B**), and the area under curve (AUC) of OGTT was evaluated (**C**). In the insulin tolerance test (ITT), the concentration of plasma glucose was measured at indicated times (**D**) and then the AUC of ITT was calculated (**E**). Western blots were used to determine the change in pAkt in the liver tissues of mice (**F**). *n* = 4–6 for each group of mice. * *p* < 0.05, ** *p* < 0.01, and *** *p* < 0.001 as compared with the Chow group or the indicated group. # *p* < 0.05 as compared with HFD groups.

**Figure 4 nutrients-15-02814-f004:**
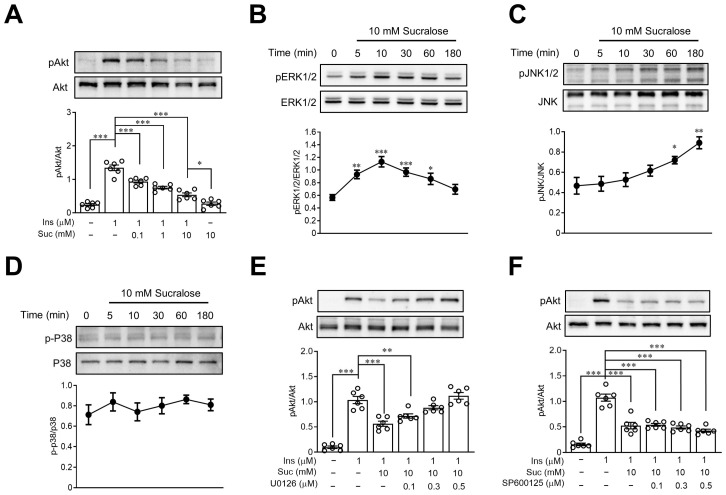
Sucralose activated ERK1/2 to disrupt insulin signaling in HepG2 cells. The protein lysates of HepG2 cells were collected to determine Akt phosphorylation by Western blots (**A**). After sucralose treatment, the protein lysates were collected to determine pERK1/2 (**B**), pJNK1/2 (**C**) and p-P38 (**D**) levels by Western blots. The cells were pretreated with indicated doses of U0126 (ERK1/2 inhibitor) (**E**) or SP600125 (JNK1/2 inhibitor) (**F**). Then the samples were treated with sucralose and insulin sequentially. The evaluation of Akt phosphorylation was performed by Western blots. *n* = 6 for each group. * *p* < 0.05, ** *p* < 0.01, and *** *p* < 0.001 as compared with 0 min (baseline) or the indicated groups.

**Figure 5 nutrients-15-02814-f005:**
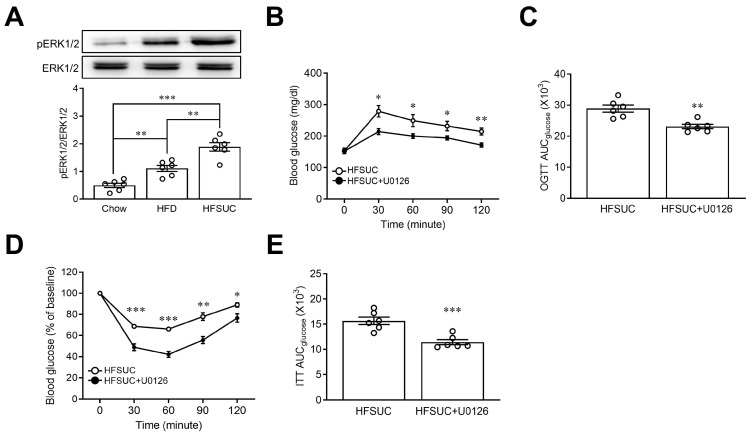
Administration of ERK1/2 inhibitor reversed sucralose-induced insulin resistance in mice. Mice were fed with chow diet (Chow), high-fat diet (HFD), or a high-fat diet supplemented with sucralose (HFSUC) for two weeks. The liver tissues were collected to determine the phosphorylation of ERK1/2 by Western blots (**A**). The HFSUC mice received daily intraperitoneal injections with U0126 (ERK1/2 inhibitor) for two weeks. Plasma glucose levels were measured at indicated time points during the oral glucose tolerance test (OGTT) (**B**), and the area under curve (AUC) of the OGTT was calculated (**C**). In the insulin tolerance test (ITT), plasma glucose concentration was measured at indicated time points (**D**), and the AUC of the ITT was also determined (**E**). *n* = 6 for each group of mice. * *p* < 0.05, ** *p* < 0.01, and *** *p* < 0.001 as compared with the indicated group or the HFSUC group.

**Figure 6 nutrients-15-02814-f006:**
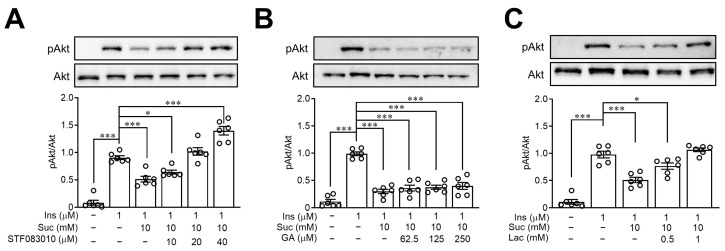
Sucralose induced insulin resistance through endoplasmic reticulum stress and sweet taste receptors. HepG2 cells were pretreated with STF083010 (endoplasmic reticulum stress inhibitor) (**A**), gymnemic acid (T1R2 inhibitor) (**B**) or lactisole (T1R3 inhibitor) (**C**) at indicated doses. Then, the samples were treated with sucralose and insulin sequentially. The Akt phosphorylation was assessed by Western blots. *n* = 6 for each group of indicated HepG2 cell group. * *p* < 0.05 and *** *p* < 0.001 as compared with the indicated groups.

## Data Availability

The supporting data for the findings of this study can be obtained from the corresponding author upon a reasonable inquiry.

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
