# Peer review of "Long-Term Consumption of Sucralose Induces Hepatic Insulin Resistance through an Extracellular Signal-Regulated Kinase 1/2-Dependent Pathway"

_nutrients, 2023, doi:10.3390/nu15122814_

Round 1
Reviewer 1 Report
Reviewer comments:
In the manuscript entitled “Long term consumption of sucralose induces hepatic insulin resistance through an extracellular signal regulated kinase 1/2-dependent pathway”, the authors investigated the effects of the sucralose consumption on hepatic insulin resistance. The authors found that the long term intake of sucralose augmented HFD-induced insulin resistance in mice, and interrupted insulin signals through a T1R3-ERK1/2-dependent pathway in the liver. The present study is of some significance for understanding the molecular adverse roles of this artificial sweeteners on glycemic homeostasis. Generally, this paper is of good quality, however, there are also some points need to be addressed before being further considered for publication.
1. In the last paragraph of the introduction part, the main works and the aims of this study can be more detailed introduced and emphasized.
2. The animal ethics statement, as well as the animal experiment approval should be mention in the 2.1 part.
3. In the 2.1 and 2.3 parts, how are the sucralose and glucose administration dosage determined? The related references are recommended to be cited in the corresponding place.
4. The resolution of the all figures in the present manuscript should be improved, and the information in the figures are not clear.
5. Line 179-180. The processing protocols for the parameter qualification are suggested moved to the materials methods part instead of the figure captions. Same suggestions for the figure 2, figure 3, figure 4 and figure 5.
6. Line 221-222. Sentence should be improved. The decrease of the pAkt protein cannot be directly explained by the impaired insulin sensitivity, more objective descriptions are recommended here, and related reference should be cited.
7. Another question is that why the authors choose the HepG2 cell model for exploring the possible underlying molecular mechanisms, instead of the primary hepatocytes? There are basic differences between them. Please give introductions in the introduction or 3.4 part.
8. As is known to all that the GLP is closely and directly connect to the insulin metabolism, however, in the 3.4 and 3.5 parts, the authors focused on the ERK signals and the ER stress, why? More detailed explanations and introduction are better provided.
9. The language of the present manuscript are generally good. However, the authors are still suggested to make double check, for some sentences are not that clear or not use scientific language.
10. The format (e.g. the line spacing and reference sequence number) of the references should be in accordance with the requirements of this journal.
The language of the present work are generally good.
Author Response
- In the last paragraph of the introduction part, the main works and the aims of this study can be more detailed introduced and emphasized.
Reply: Thank you for your valuable comment. The related information has been provided in our revised manuscript (Page 2, Line 91-96).
- The animal ethics statement, as well as the animal experiment approval should be mention in the 2.1 part.
Reply: The related information has been provided in the 2.1 part of our revised manuscript, according to your valuable suggestion.
- In the 2.1 and 2.3 parts, how are the sucralose and glucose administration dosage determined? The related references are recommended to be cited in the corresponding place.
Reply: Since glucose tolerance test using 1 g/ kg glucose is a common strategy for the evaluation of glucose utility [1, 2], we used the same dosage of sucralose to investigate the effects of short-term administration of sucralose on glucose homeostasis. In addition, we followed a previous study the effects of long-term administration of sucralose that the dosage of sucralose was converted from FDA approved acceptable daily intake dose for animal study [3, 4]. The related information has been provided in our revised manuscript (Page 3, Line 112-116, and 130).
The dose of sucralose was and the reference was attached in the paragraph, please see line 112-116. As for glucose dosage in the 2.3 parts, the corresponding reference has been added at line 130.
Reference
- Alquier, Thierry, and Vincent Poitout. "Considerations and Guidelines for Mouse Metabolic Phenotyping in Diabetes Research." Diabetologia 61, no. 3 (2018): 526-38.
- Nagy, C., and E. Einwallner. "Study of in Vivo Glucose Metabolism in High-Fat Diet-Fed Mice Using Oral Glucose Tolerance Test (Ogtt) and Insulin Tolerance Test (Itt)." J Vis Exp, no. 131 (2018).
- Food, US, and Drug Administration. "Food Additives Permitted for Direct Addition to Food for Human Consumption; Sucralose." Fed Reg 64 (1999): 43908-09.
- Reagan-Shaw, S., M. Nihal, and N. Ahmad. "Dose Translation from Animal to Human Studies Revisited." FASEB Journal 22, no. 3 (2008): 659-61.
- The resolution of the all figures in the present manuscript should be improved, and the information in the figures are not clear.
Reply: The resolution of each figure and the detail information has been provided in the revised figure legends.
- Line 179-180. The processing protocols for the parameter qualification are suggested moved to the materials methods part instead of the figure captions. Same suggestions for the figure 2, figure 3, figure 4 and figure 5.
Reply: Our manuscript has been revised following your valuable suggestions.
- Line 221-222. Sentence should be improved. The decrease of the pAkt protein cannot be directly explained by the impaired insulin sensitivity, more objective descriptions are recommended here, and related reference should be cited.
Reply: Thank you for your valuable comment. The sentences have been revised and the corresponding reference is also added in our revised manuscript (Page 6, Line 238-240).
- Another question is that why the authors choose the HepG2 cell model for exploring the possible underlying molecular mechanisms, instead of the primary hepatocytes? There are basic differences between them. Please give introductions in the introduction or 3.4 part.
Reply: In order to saving lives of animals, cell lines are commonly used for exploring the possible underlying molecular mechanisms. In addition, HepG2 cell line is capable of expressing glycogen synthesis, insulin signaling and lipid metabolism [1], and previous studies also used this cell line to investigate the effects of sucralose on liver. The introduction and corresponding reference were added in our revised manuscript (Page 7, Line 257-259).
Reference
- Donato, M. T., L. Tolosa, and M. J. Gómez-Lechón. "Culture and Functional Characterization of Human Hepatoma Hepg2 Cells." Methods in Molecular Biology 1250 (2015): 77-93.
- As is known to all that the GLP is closely and directly connect to the insulin metabolism, however, in the 3.4 and 3.5 parts, the authors focused on the ERK signals and the ER stress, why? More detailed explanations and introduction are better provided.
Reply: Thank you for your valuable comments. In the present study, we found opposite effects of sucralose after short-term and long-tern treatment of sucralose in mice. Since it was known that sucralose increases ER stress and activates ERK1/2 activity and activation of these signals contributes to the development of insulin resistance, we therefore speculated that sucralose may induce insulin resistance through the ERK signals and the ER stress [1, 2]. (Please see page 6, line 261-264 and page 7, line 289-290)
Reference
- Ozaki, K. I., M. Awazu, M. Tamiya, Y. Iwasaki, A. Harada, S. Kugisaki, S. Tanimura, and M. Kohno. "Targeting the Erk Signaling Pathway as a Potential Treatment for Insulin Resistance and Type 2 Diabetes." American Journal of Physiology: Endocrinology and Metabolism 310, no. 8 (2016): E643-e51.
- Ajoolabady, A., S. Liu, D. J. Klionsky, G. Y. H. Lip, J. Tuomilehto, S. Kavalakatt, D. M. Pereira, A. Samali, and J. Ren. "Er Stress in Obesity Pathogenesis and Management." Trends in Pharmacological Sciences 43, no. 2 (2022): 97-109.
- The language of the present manuscript are generally good. However, the authors are still suggested to make double check, for some sentences are not that clear or not use scientific language.
Reply: We had invited native English speaker to edit our manuscript. Thank you very much.
- The format (e.g., the line spacing and reference sequence number) of the references should be in accordance with the requirements of this journal.
Reply: Our manuscript has been revised in accordance with the requirements of this journal.

Reviewer 2 Report
In this review, Tsai et al. evaluated the influence of long-term consumption of sucralose on the development of hepatic insulin resistance. The authors proposed an ERK1/2 dependent mechanism by which sucralose impairs hepatic insulin sensitivity. The authors provide in vivo and in vitro evidence to support their hypothesis. It is a interesting study with few major concerns to be addressed.
1. Figure 4 and Figure 5 are missing.
2. The authors stated that sucralose induced ERK1/2 phosphorylation in HepG2 cells, how about the effect of long-term sucralose consumption on pERK1/2 levels in the liver?
3. Following the above, the authors should provide in vivo evidence to support that ERK1/2 mediates sucralose-induced hepatic insulin resistance. For example, whether ERK1/2 inhibitor could improve insulin sensitivity in long-term sucralose treated mice.
Author Response
- Figure 4 and Figure 5 are missing.
Reply: Thank you for your valuable comments. The missing figures have been provided in our revised manuscript.
- The authors stated that sucralose induced ERK1/2 phosphorylation in HepG2 cells, how about the effect of long-term sucralose consumption on pERK1/2 levels in the liver?
Reply: In the present study, we found that long-term administration of sucralose also increased the pERK1/2 levels in the liver. The related information has been provided in our revised manuscript. (Page 7, Line 270-273 and Figure 4G)
- Following the above, the authors should provide in vivo evidence to support that ERK1/2 mediates sucralose-induced hepatic insulin resistance. For example, whether ERK1/2 inhibitor could improve insulin sensitivity in long-term sucralose treated mice.
Reply: It was known that the activity of ERK1/2 plays a crucial role in insulin sensitivity. knockout of the ERK gene demonstrated enhanced systemic insulin sensitivity [1]. In addition, the use of U0126, an inhibitor of ERK has shown to improve blood glucose levels in ob/ob diabetic mice [2]. Additionally, the administration of PD184352, a MEK inhibitor that blocks the ERK pathway, has been found to improve diabetes in db/db and high-fat diet-fed KKAy mice models [3]. Since the ERK1/2 inhibitors themselves exhibited glucose-lowing effects in animals, we therefore used HepG2 cells to investigated the possible molecular mechanisms of sucralose-induced insulin resistance.
Reference
- Jiao, P., B. Feng, Y. Li, Q. He, and H. Xu. "Hepatic Erk Activity Plays a Role in Energy Metabolism." Molecular and Cellular Endocrinology 375, no. 1-2 (2013): 157-66.
- Hwang, S. L., Y. T. Jeong, X. Li, Y. D. Kim, Y. Lu, Y. C. Chang, I. K. Lee, and H. W. Chang. "Inhibitory Cross-Talk between the Ampk and Erk Pathways Mediates Endoplasmic Reticulum Stress-Induced Insulin Resistance in Skeletal Muscle." British Journal of Pharmacology 169, no. 1 (2013): 69-81.
- Ozaki, K. I., M. Awazu, M. Tamiya, Y. Iwasaki, A. Harada, S. Kugisaki, S. Tanimura, and M. Kohno. "Targeting the Erk Signaling Pathway as a Potential Treatment for Insulin Resistance and Type 2 Diabetes." American Journal of Physiology: Endocrinology and Metabolism 310, no. 8 (2016): E643-e51.

Reviewer 3 Report
The study evaluate the short and long term administration of sucralose on the glucose metabolism, as well as the mechanisms which makes the prolonged consumption of this sweetener to induce insulin resistance in animal and cell models.
Introduction
line 81 add zebra fish in brackets after the Latin name
Line 83-90 it will be better to specify that the species tested were mice, because it became a little confusing after the mentioning of the zebra fish
Materials and Methods
Animals
What is the number of animals in each group?
Results
All Figures looks blurry. It will be better if they are a little bit clearer. Also, the labels of Fig 4 and 5 are too small and very difficult to see. The authors may also consider a different way of presenting the data, in order to be easier on the readers to understand it.
References
Reduce the space between the cited papers
Author Response
Manuscript Number: nutrients-2440446
Authors: Meng-Jie Tsai, Chung-Hao Li, Hung-Tsung Wu, Hsin-Yu Kuo, Chung-Teng
Wang, Hsiu-Ling Pai, Chih-Jen Chang, Horng-Yih Ou
Title: Long term consumption of sucralose induces hepatic insulin resistance through an extracellular signal regulated kinase 1/2-dependent pathway
Dear Editor,
The current revision of our submitted manuscript has been amended accordance with your comments. The point-by-point replies for each comment are indicated in the file showing the reply to you. Changes of the revised part were marked in red. The figures are also presented according to your valuable requirements for publication. We sincerely hope that this revised version of our submitted manuscript will be suitable to meet your excellent standards of acceptance. Thanks for your kind consideration.
Best regards,
Prof. Horng-Yih Ou, M.D., Ph.D.
The corresponding author
Introduction
- Line 81 add zebra fish in brackets after the Latin name.
Reply: Thank you for your valuable comments. Our manuscript has been revised according to your suggestion. (Page 2, Line 80)
- Line 83-90 it will be better to specify that the species tested were mice, because it became a little confusing after the mentioning of the zebra fish.
Reply: Our manuscript has been revised according to your valuable suggestion. Thank you. (Page 2, Line 82)
Materials and Methods
- Animals -What is the number of animals in each group?
Reply: The sample size of each group of the mice was provide in the figure legends in our revised manuscript.
Results
- All Figures looks blurry. It will be better if they are a little bit clearer. Also, the labels of Fig 4 and 5 are too small and very difficult to see. The authors may also consider a different way of presenting the data, in order to be easier on the readers to understand it.
Reply: The resolution of each figure and the detail information has been provided in the revised figure legends.
References
- Reduce the space between the cited papers
Reply: Our manuscript has been revised in accordance with the requirements of this journal.

Round 2
Reviewer 1 Report
I have no more questions regarding this manuscript, and I think this good paper can be accepted in the present form.
Author Response
Thank you for your valuable comments.
Reviewer 2 Report
the authors addressed all my concerns
Author Response
Thank you for your valuable comments.